# REFORMULATING DOMAIN ADAPTATION OF LARGE LANGUAGE MODELS AS ADAPT-RETRIEVE-REVISE

## ABSTRACT

While large language models (LLMs) like GPT-4 have recently demonstrated astonishing zero-shot capabilities in general domain tasks, they often generate content with hallucinations in specific domains such as Chinese law, hindering their application in these areas. This is typically due to the absence of training data that encompasses such a specific domain, preventing GPT-4 from acquiring in-domain knowledge. A pressing challenge is that it's not plausible to continue training LLMs of such scale on in-domain data.

This paper introduces a simple and effective domain adaptation framework for GPT-4 by reformulating generation as an **adapt-retrieve-revise** process. The initial step is to **adapt** an affordable 7B LLM to the target domain by continuing learning on in-domain data. When solving a task, we leverage the adapted LLM to generate a draft answer given a task query. Then, the draft answer will be used to **retrieve** supporting evidence candidates from an external in-domain knowledge base. Finally, the draft answer and retrieved evidence are concatenated into a whole prompt to let GPT-4 assess the evidence and **revise** the draft answer to generate the final answer.

Our proposal combines the advantages of the efficiency of adapting a smaller 7B model with the evidence-assessing capability of GPT-4 and effectively prevents GPT-4 from generating hallucinatory content. In the zero-shot setting of four Chinese legal tasks, our method improves accuracy by 33.3% compared to the direct generation by GPT-4. When compared to two stronger retrieval-based baselines, our method outperforms them by 15.4% and 23.9%. Our code will be released [1].

## 1 INTRODUCTIONS

Recent large language models (e.g., GPT-4) bring remarkable improvements in various general domain NLP tasks (Brown et al., 2020a; OpenAI, 2023; Thoppilan et al., 2022; Chowdhery et al., 2022; Rae et al., 2022; Hoffmann et al., 2022). However, in specific domains such as the Chinese legal domain, the performance of such general LLMs still lags considerably behind. We show a real example of Chinese LegalQA (Chen et al., 2023) on the left of Figure 1, which requires the model to generate the corresponding legal provision (i.e., the law name and the clause index) and the rationale for the judgment, given a brief case description as the query.

We initialize the research with a preliminary examination of utilizing GPT-4 to address the Chinese LegalQA task, which involves responding with a law clause relevant to a given query case. Figure 1 reveals the extremely low recall of directly prompting the query case to ask GPT-4 to generate the corresponding law clause. Though the generated answers are grammatically fluent, they often consist of non-logical content, factual mistakes, and fail to refer to the correct legal provision (also known as "hallucination"). For example, in Figure 2, the direct generation of GPT-4 seems to be logical but actually has no clues in the Chinese laws. A potential reason is the insufficient Chinese legal domain text during pretraining, leading to a lack of domain knowledge acquisition, and consequently generating hallucinatory content.

For the LLMs with the scale of GPT-4, it's generally not feasible for researchers to do learning-based adaptation. The enormous model size could make the cost of continual learning extremely high, and

---

[1]An anonymous link. The domain adapted 7B LLM will be also released.

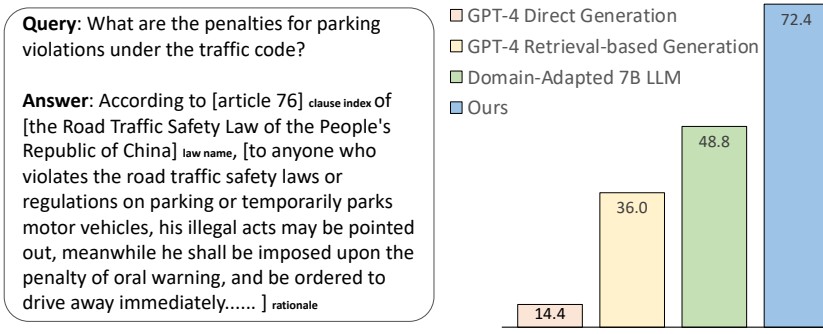

Figure 1: **Left**: A real example of Chinese LegalQA. The square brackets and subscripts are offered for the purpose of clear demonstration, not actually exist in the ground-truth answer or generation. **Right**: The recall on the LegalQA dataset.

meanwhile, the access functions are often limited by APIs. Therefore, recent work (Lewis et al., 2020b) introduces retrieval-based methods that first use the given query to retrieve relevant evidence candidates from the external domain-specific knowledge base or the internet and then concatenate the query and the evidence candidates into the prompt. GPT-4 could implicitly validate the relevance between the query and the evidence, as well as the correctness of the evidence, before producing a generation. In our replicated results, the retrieval-based method improves the LegalQA recall from 14.4% (direct generation) to 36.0%. This suggests that even though GPT-4 may not generate domain content, it possesses sufficient evidence-assessing capacity to select the correct evidence from candidates. Nevertheless, the retrieval module is limited by the capability of representation mapping from query to evidence and is also influenced by the domain issue, leading to a decline in search quality. GPT-4 still produces hallucinations in responses as the middle answer in Figure 2.

On the other hand, with the rapid development of open LLMs led by LLaMA (Touvron et al., 2023), it becomes affordable to continually train an open LLM tailored to your demands on sufficient in-domain texts, resulting in a domain-adapted LLM. We therefore conduct the second examing of continually training Baichuan 7B (Baichuan-inc, 2023), a Chinese foundation model, on over 50B Chinese legal data. Its performance significantly outperforms GPT-4 and even surpasses the retrieval-based GPT-4 generation on Chinese LegalQA. Hallucinations caused by the lack of domain knowledge are largely reduced but not completely solved. As shown in the right answer in Figure 2, adapted LLM generates generally correct responses but still makes errors in certain words. Although the law name is correct and the rationale part is reasonable, the clause index is a hallucinatory generation which raises the difficulty in anchoring target clauses. We argue that these fails are accordingly caused by the limited capability of a 7B size to understand the query and the evidence.

Building upon the observation of the evidence-assessing capability of GPT-4 and the high-quality domain content generated by the domain-adapted 7B model, this paper proposes a novel approach to reformulate GPT-4's domain content generation to an adapt-retrieve-revise process: (1) the domain-adapted model generates a draft answer given a query; (2) the retrieval module uses the draft answer as input for searching external evidence candidates because the answer is usually more informative and semantically similar to the evidence compared to the query as long as the answer quality is acceptable; (3) GPT-4 assesses retrieved evidence and revises the draft answer to generate the final answer.

The rest sections of the paper anchor the Chinese legal domain and comprehensively validate the effectiveness of our proposal. In Section 2, we explain each stage of our adapt-retrieve-revise method and elaborate on the implementation details. In Section 3 and 4, we conduct the experiments and the result analysis on four Chinese legal domain tasks. The experimental results show that our proposal significantly improves the direct generation and the retrieval-based generation baselines. In the final Section 6, we elicit the conclusion and future work. To the best of our knowledge, this is the first study to examine the zero-shot performance of LLMs on four Chinese legal benchmarks.

> **Query**: What are the penalties for parking violations under the traffic code?
> **Answer**:
>
> | **GPT-4's Direct Generation**: | **GPT-4's Retrieval-based Generation**: | **Domain-adapted LLM's Generation**: |
> |---|---|---|
> | In China, the penalties for parking violations may vary depending on the specific area and circumstances. Typically, parking penalties may include fines, demerit points and, in severe cases... | According to [article 37] of [the Road Traffic Safety Law of the People's Republic of China], [where a special driveway is delimited on a road, only prescribed…] | According to [article 90] of [the Road Traffic Safety Law of the People's Republic of China], [to anyone who violates the road traffic safety laws or regulations on parking or parks...] |

Figure 2: Examples of hallucinations of various models. Red denotes the content containing hallucinations. The ground-truth answer is shown in Figure 1

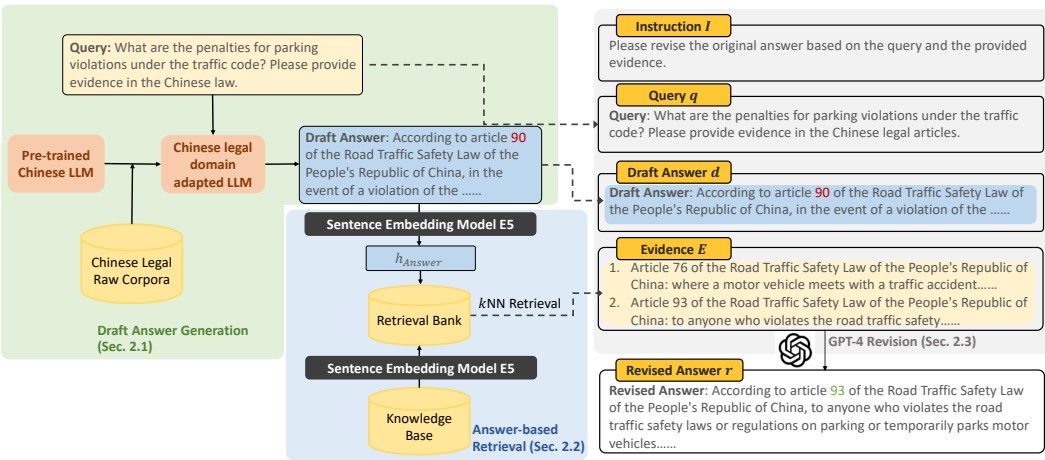

Figure 3: **Overview of our proposed method.**

## 2 METHODOLOGY

Our adapt-retrieve-revise method consists of three steps. In the first step (Section 2.1), we continually train a Chinese pre-trained LLM on the Chinese legal domain corpora to derive a domain-adapted legal LLM and given the query, the legal LLM will generate the draft answer. In the second step (Section 2.2), we use a sentence embedding model to produce embeddings for both the draft answer and each paragraph in the corresponding knowledge base, then evidence retrieval will be computed by the similarities between the answer embedding and the paragraph embeddings. In the third step (Section 2.3), we concatenate the query, the draft answer, and the retrieved evidence in the prompt for GPT-4 to revise and produce the final response. Figure 3 shows the overview of our method. In the following sections, we will introduce details of each step.

### 2.1 GENERATE THE DRAFT ANSWER BY THE DOMAIN-ADAPTED LLM

Recently, various Chinese pre-trained LLMs have occurred, such as Baichuan 7B, Qwen 7B (Alibaba-inc, 2023), etc. In this paper, we chose Baichuan-7B as the base model for our proposal. For the Chinese legal raw corpora, we build the continual learning data from two open sources:

- **Chinese law clauses** (`https://flk.npc.gov.cn/`) form the foundation of the judicial system, containing a wealth of legal terms, provisions, and judicial practices. They are essential for the model to understand and generate relevant content.

- **Chinese judgments online** (https://wenshu.court.gov.cn/) is the largest online publication platform for legal documents in China. The platform contains judicial documents from courts at all levels, covering various legal fields such as civil, criminal, administrative, and enforcement. We believe these documents contain knowledge for LLMs to understand the usage of laws in various scenarios, and we collect 100M documents from this platform as the training data.

In total, we trained 50B tokens with the input length limit of 16,384 and the batch size of 256 on 32 A100 GPUs, and the time-consuming is 167 hours. After continual learning, we subsequently supervised fine-tuning our model on 70K instruction examples, including 52K GPT-4 self-instruct Chinese data (Peng et al., 2023) and 18K legal instructions (See Appendix A.1) for the alignment.

Then, given an input query, we will first prompt the trained 7B legal LLM to generate the draft answer, which will be used in the next step. For the prompt, we add the instruction "Please provide evidence in the Chinese law" at the end of the query to enforce the model to generate related law clauses, as in Figure 3.

## 2.2 ANSWER-BASED EVIDENCE RETRIEVAL

Since the draft answer of the 7B legal LLM is usually more informative and semantically similar to the evidence than the query. We further use the generated evidence to retrieve ground-truth evidence from the target knowledge base for the purpose of revision since it contains much more information than the query, even though the hallucinations can not be totally reduced. We implement this method with two subsequent steps: knowledge bank construction and retrieval.

**Knowledge Bank Construction**     For the $i$-th paragraph $p_i$, we construct the *key-value* pair $(\boldsymbol{p}_i, p_i)$ where the *key* $\boldsymbol{p}_i$ is the representation obtained from the sentence embedding model and the *value* $p_i$ denotes the paragraph. In our experiment, we utilize Multilingual-E5-large (Wang et al., 2022), a Roberta-based (Liu et al., 2019) sentence embedding model that achieves robust performance on various tasks. The memory $(\mathcal{K}, \mathcal{V}) = \{(\boldsymbol{p}_i, p_i) | p_i \in \mathcal{KB}\}$ is thus the set of all *key-value* pairs constructed from all the paragraphs in the external knowledge base $\mathcal{KB}$.

**Retrieval**     Given the generated draft answer $d$, the sentence embedding model E5 outputs its representation $\boldsymbol{h}_{Answer}$. We then query the constructed knowledge bank with $\boldsymbol{h}_{Answer}$ to retrieve its $k$ nearest neighbors $E$ according to a distance function by $L^2$ distance.

## 2.3 GPT-4 REVISION

To effectively combine the high-quality draft answers generated by the 7B domain adapted model with GPT-4's powerful evidence-assessing capability, we propose the following process. As shown in Figure 3, the whole prompt consists of the following components: (1) the instruction $I$ to require GPT-4 to revise the draft answer given the query and the evidence candidates; (2) the query $q$ itself; (3) the draft answer $d$ for GPT-4 to revise; (4) and the retrieved evidence candidates $E$ to provide related Chinese legal knowledge for GPT-4. Then, the final revised answer $r$ will be outputted by $GPT4(I, q, d, E)$.

## 3 EXPERIMENTS

We conducted a series of experiments to compare our adapt-retrieve-revise method to the baselines of direct generation and retrieval-based generation on various Chinese legal benchmarks. We find that our proposal robustly improves performance on all the tasks. We show the task settings and experimental results in this section.

## 3.1 CHINESE LEGAL DOMAIN TASKS

We evaluated our Adapt-Retrieve-Revise method on a diversity of tasks in the zero-shot setting, and we divided these tasks into three categories by the legal knowledge base for retrieval:

Table 1: **Main Results on four Chinese legal datasets**.

| Models | Retrieval | Revisor | Chinese law articles | | Textbooks | Avg. |
| | | | LCR | CP | LegalQA | JEC-QA | |
|---|---|---|---|---|---|---|---|
| Direct Generation | | | | | | | |
| GPT-4 | - | - | 67.6 | 71.2 | 14.4 | 36.2 | 47.4 |
| 7B legal LLM | - | - | 88.4 | 84.0 | 48.8 | 39.8 | 65.3 |
| Retrieval-based Generation | | | | | | | |
| GPT-4 | Query-based | - | 74.4 | 75.2 | 36.0 | 41.6 | 56.8 |
| 7B legal LLM | Query-based | - | 87.6 | 82.4 | 50.2 | 40.8 | 65.3 |
| Adapt-Retrieve-Revise | | | | | | | |
| 7B legal LLM | Answer-based | 7B legal LLM | 88.4 | 83.6 | 49.0 | 40.2 | 65.3 |
| 7B legal LLM (ours) | Answer-based | GPT-4 | **96.4** | **87.8** | **72.4** | **66.2** | **80.7** |

- **Law CLause Recommendation (LCR) and Criminal Prediction (CP)** (Xiao et al., 2018) are two tasks using the legal report as the input, and let the model generate the most related law clause and predict the criminal type based on the law clause. For these two tasks, we use the **Chinese law clauses** as the domain knowledge base for retrieval.

- **LegalQA** is a filtered set of EUQALS (Chen et al., 2023) that, given an input query, the model should generate an answer based on the most related legal clause. The filtering is based on the quality of the questions and we will release the filtered set. We also use the **Chinese law clauses** as the domain knowledge base for retrieval.

- **JEC-QA** (Zhong et al., 2020) is the official test for getting a lawyer's certificate in China. We chose the single-choice selection questions in our evaluations with the **Legal Textbooks** (`https://github.com/thunlp/jec-qa`) as the knowledge base for retrieval.

- **Similar Case Retrieval** (Ma et al., 2021) is the task that, given a query legal scenario as the input, we aim at selecting similar **Legal Judgement Documents** from the 100 candidates. We conducted this experiment to assess the reliability of our proposed retrieval method in Section 4.1.2.

Due to the cost of GPT-4 API and the human evaluation, we randomly sampled a subset of 250 test examples for each task of LCR, CP, LegalQA, and JEC-QA.

## 3.2 EVALUATION METRICS

Since generative models produce diverse formats in the output and the Chinese legal domain has its own features, checking the evaluation metrics in the experiments is crucial. For tasks LCR, CP, and LegalQA, our metric is the *recall* of whether the title of the ground-truth law clause is included in the generated answer. This is because, in real-world applications, with the correct title, the contents of the law clause can be easily revised by the rule-based system, indicating that the title is more important than the content.

For the JEC-QA task, we use accuracy as the metric, but controlling the output into an identical format for automatic evaluation is difficult, especially for the 7B LLM that has not been fine-tuned on the JEC-QA task. We select human evaluation to ensure the accuracy of our evaluation.

For the Similar Case Retrieval task, we chose the widely used *precision@k* and *MAP* as the evaluation metrics.

## 3.3 MAIN RESULTS

We provide the main results as in Table 1. Generally, we compare our proposed method with direct generations and retrieval-based generations using the query, showing that our method outperforms all baselines by a large margin. Our main results also provide some ablation results.

First, our 7B legal LLM significantly beats GPT-4, and even compared with the retrieval-based generation of GPT-4, the 7B legal LLM still outperforms on three tasks and has competitive results

on the JEC-QA task, indicating that our continual learning on Chinese legal raw corpora shows a fast and effective domain adaptation on various legal tasks.

Then, considering the results of GPT-4 and the GPT-4 retrieval-based generation, we find that after providing evidence of related legal knowledge, GPT-4 can improve its responses significantly (+9.92%). This indicates that the retrieval-based method is a proper way to reduce hallucinations caused by the lack of domain knowledge, and owing to the robust evidence-assessing capacity, GPT-4 can adapt to the Chinese legal domain well with convincing evidence available.

In our final experiment, using the draft answers generated by the 7B legal model for retrieval and revision, the performance significantly exceeded two query-based retrieval baselines by large margins of 15.4% and 23.9%. It's worth noting that the improvement here comes from both the enhanced answer-based retrieval quality and the revision setup. In the subsequent ablation study, we will further examine the quantified improvement of the retrieval quality through an additional retrieving task.

An interesting observation is that, by comparing the direct generation of the 7B legal model and the adapt-retrieve-revise method with the revision model as the legal 7B model, we find that with retrieved evidence, the revised answers seem to be no obvious difference from the direct generation. This indicates that the 7B legal LLM shows almost zero evidence-assessing capacity.

## 4 ABLATION STUDIES & FURTHER ANALYSIS

### 4.1 ANALYSIS OF RETRIEVAL METHODS

We believe the answer-based approach is more effective due to two reasons. (1) The query-based retrieval requires a query-to-evidence representation mapping. The answers are usually more semantically similar to the evidence, which avoids the mapping process. (2) A query is often very brief, while an answer containing the legal provision and rationale is more informative. In this subsection, we analyze the retrieval component, including the apple-to-apple comparisons between the query-based and answer-based performance and the influence of answer quality for the answer-based retrieval.

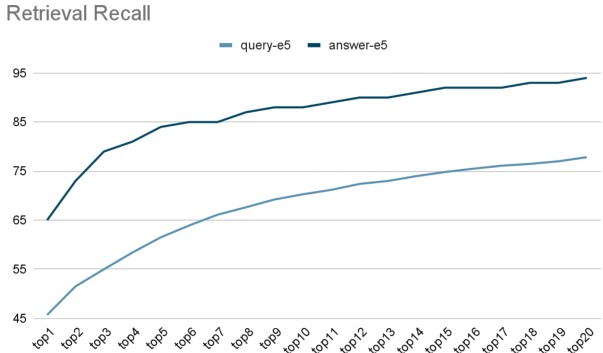

Figure 4: **Comparison of retrieval recalls on the LegalQA dataset.**

### 4.1.1 RETRIEVING A QUERY OR RETRIEVING AN ANSWER?

We ordered the top-similar law clauses in each retrieval and evaluated the recall in top-$k$, indicating whether the ground-truth law clause appears in the top-$k$ retrieved law clauses. As shown in Figure 4, the top-1 retrieved law clause based on the answer competes with the top-5 law clauses based on the query, and the answer-based retrieval beats the query-based retrieval with a large margin for all $k$. This confirms our first reason that the draft answer contains much more information than the query for retrieval, indicating that LLMs can be intrinsic retrievers.

We further compare the query-based and answer-based retrieval on a public **Similar Case Retrieval** task. This task aims to select similar legal judgments based on the query from the candidates with a

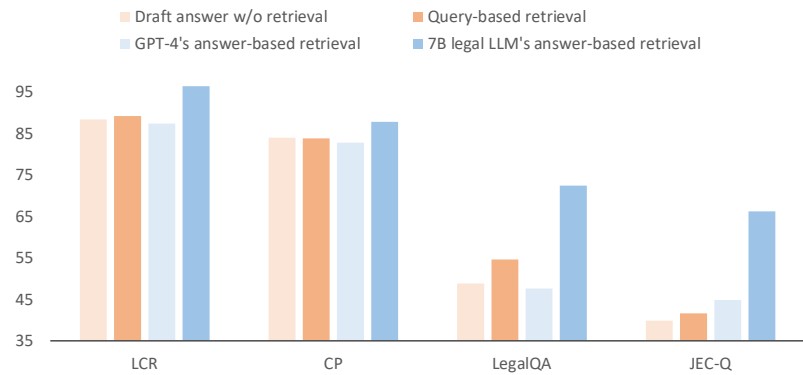

Figure 5: We compare performances of the draft answer of 7B legal LLM and our proposed adapt-retrieve-revise model using different contents in retrieval.

Table 2: **Results of two retrieval setups on the Similar Case Retrieval dataset**.

| Setup | Precision@5 | Precision@10 | MAP |
|---|---|---|---|
| Query-based | 42.08 | 41.95 | 47.8 |
| Answer-based | **45.19** (+3.11) | **42.08** (+0.13) | **49.49** (+1.69) |

query the case brief given. As shown in Table 2, we compare two setups: 1) using the original query to retrieve, 2) using the legal 7B LLM to complete a whole legal judgment document given the brief query, and then retrieving. We follow the original task repository for the other settings: `https://github.com/myx666/LeCaRD`. The results show that on each metric, the answer-based retrieval works better, indicating that using the generated answer provides a more robust retrieval.

### 4.1.2 DOES THE QUALITY OF ANSWER MATTER FOR ANSWER-BASED RETRIEVAL?

It's an intuitive thought that the quality of answers will significantly impact the outcome of answer-based retrieval. Therefore, we compare the retrieval using the answers of GPT-4 and the 7B legal LLM. We change the contents in retrieval for our proposed adapt-retrieve-revise method. As shown in Figure 5, by comparing query-based and GPT-4's answer-based retrievals, we find that the answer-based retrieval fails on three datasets (LCR, CP, LegalQA), indicating that the lack of domain knowledge in the GPT-4 responses leads to a more noisy retrieval, which even hurts the performance of the draft answer (LCR, CP, LegalQA). Meanwhile, after domain adapting, our 7B legal LLM provides robust answers in retrieval and leads to the best performances, indicating that the learned Chinese legal domain knowledge improves our answer-based retrieval.

### 4.2 CASE ANALYSIS OF THE IMPROVEMENTS AFTER THE GPT-4 REVISION

We conclude the improvements by GPT-4 in three aspects as shown in Figure 6:

- **Adding law clauses for reference**: Sometimes, the 7B legal LLM only provides a fluent response without following the input instructions to provide the key information of the referred law name and clause index. In this case, the faithfulness of the answer remains unchecked for the users. However, after the revision, each answer is equipped with the referred law clause, which makes it easier to check the accuracy of the responses.

- **Revise hallucinations in the evidence**: even the domain-adapted LLM can provide evidence from its learned legal knowledge; the hallucination remains to some degree, such as the wrong clause index, even the law name and rationale are roughly correct. Since the rationale content is accurate, the answer-based retrieval will search for the correct evidence, and the revision by GPT-4 will solve the hallucination to produce a more robust response.

**Query**: What are the provisions regarding the scope of legal succession and the order of succession?

**Draft answer**:
1. The spouse, children and parents of the decedent are the first order of heirs; 2. The siblings, grandparents and grandparents of the decedent are the second order of heirs…

**Revised answer**:
According to article 1127 of Civil Code of the People's Republic of China, 1. The spouse, children and parents of the decedent are the first order of heirs; 2. The siblings, grandparents and…

**Query**: What are the penalties for parking violations under the traffic code?

**Draft answer**:
According to article 90 of the Road Traffic Safety Law of the People's Republic of China, to anyone who violates the road traffic safety laws or regulations on parking or parks motor vehicles…

**Revised answer**:
According to article 93 of the Road Traffic Safety Law of the People's Republic of China, to anyone who violates the road traffic safety laws or regulations on parking or temporarily parks motor vehicles…

**Query**: Which traffic accidents can be handled according to the simplified procedure?

**Draft answer**:
According to article 74 of the Road Traffic Safety Law of the People's Republic of China, with respect to a dispute over indemnity for the damage from a traffic accident, the parties concerned…

**Revised answer**:
According to article 70 of the Road Traffic Safety Law of the People's Republic of China, If a traffic accident occurs on road, the vehicle driver shall immediately park the vehicle and protect the scene…

Figure 6: **Case studies on the GPT-4 revision**.

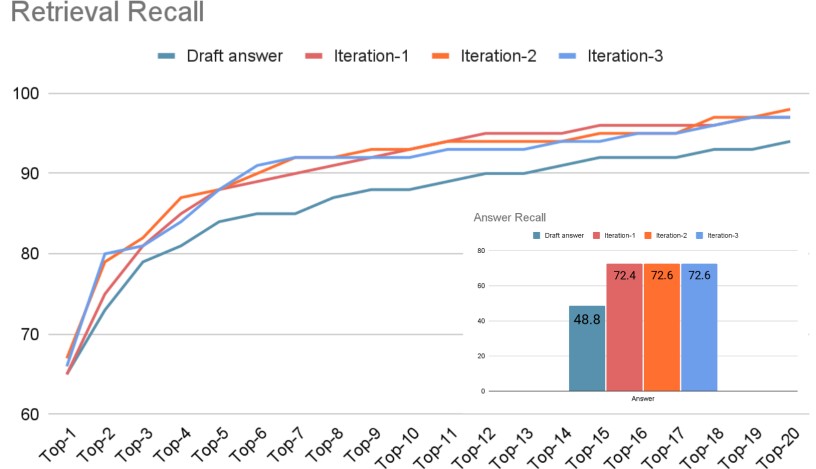

Figure 7: **Comparison of iterations on LegalQA dataset.**

- **Choose correct evidence**: In a more significant scenario, even though the 7B legal model's answers might contain partial hallucinatory content, the retrieval component can still possibly identify correct evidence through the partially correct descriptions in the rationale generation. During the revision stage, GPT-4 could assess the correct evidence, leading to the generation of correct answers.

### 4.3 DOES THE ITERATION MAKE THE GENERATION BETTER?

Since our method provides more accurate responses than the original response from the domain LLM, one question is whether this procedure can be iterated to improve the responses. We can use the revised response to retrieve related evidence and further improve the response. To verify this probability, we iteratively test on the LegalQA dataset. As the result is shown in Figure 7, during the iteration, the retrieval recall does not show consistent improvements compared with the first revision, leading to the performance nearly unchanged.

## 5    RELATED WORK

### 5.1    TASKS IN THE CHINESE LEGAL DOMAIN

The rapid advancements in LLMs have significantly impacted various domains, including the legal industry. This gives rise to the occurrence of legal datasets, such as the Challenge of AI in Law (CAIL, `http://cail.cipsc.org.cn/index.html`), LeCaRD (Ma et al., 2021), JEC-QA (Zhong et al., 2020) and EQUALS (Chen et al., 2023). These datasets cover document classification, summarization, question answering, information extraction, similar document retrieval, and other popular NLP tasks in the Chinese legal domain. To the best of our knowledge, this paper is the first work to exam the zero-shot performances on these legal datasets.

### 5.2    CHINESE LEGAL LLMS

As for Chinese legal LLMs, recent work utilizes a paradigm of continual learning in the legal domain, and a substantial number of instruction fine-tuning datasets were constructed to augment the proficiency in rendering legal advice. Particularly, the series of LaWGPT (Song, 2021) has been developed by leveraging foundational models such as Chinese-LLaMA-7B (Cui et al., 2023b), Chat-GLM (Du et al., 2022), and Chinese-alpaca-plus-7B (Cui et al., 2023b). Lawyer LLaMa (Huang et al., 2023) base on the more advanced Chinese-LLaMa-13B (Cui et al., 2023b), On the other hand, LexiLaw (Hai, 2023), built on the foundation of ChatGLM-6B (Du et al., 2022), underwent training through the application of three distinct methods, namely LoRA (Hu et al., 2022), P-tuning (Liu et al., 2021), and fine-tuning. Furthermore, Chatlaw (Cui et al., 2023a) received training based on both Ziya-LLaMA-13B-v1 (IDEA-CCNL, 2023) and Anima-33B (lyogavin, 2023). DISC-LawLLM Yue et al. (2023) adopted legal syllogism prompting strategies to construct supervised fine-tuning datasets and fine-tune LLMs with legal reasoning capability. A primary reason hindering us from utilizing such existing models is that they have often been trained on those publicly legal tasks already. Therefore the zero-shot capabilities can not be truly reflected. We thus continue training the general Baichuan 7B on legal data by ourselves.

### 5.3    RETRIEVAL-AUGMENTED INFERENCE

In scenarios where language models are confronted with tasks necessitating an infusion of external knowledge, a retriever mechanism can be used to provide evidence. The Retrieval-Augmented Generation (RAG) (Lewis et al., 2020b) system incorporates a BERT-based (Devlin et al., 2019) Document Retrieval Process (DRP) and utilizes BART (Lewis et al., 2020a) for answer generation. Analogously, the EMDR2 (Yu et al., 2023) employs the expectation-maximization algorithm to account for multiple retrieved documents. The Atlas (Izacard et al., 2022) builds upon the EMDR2 framework, and by synergistically training the retriever and reader components, it demonstrates few-shot learning capabilities commensurate with the 540B PalM (Chowdhery et al., 2022). RETRO (Borgeaud et al., 2022) benefits from retrieval mechanisms on expansive corpora during its pre-training phase and exhibits performance in close alignment with those of GPT-3 (Brown et al., 2020b).

## 6    CONCLUSIONS AND FUTURE DISCUSSIONS

In this paper, we reformulate the zero-shot domain content generation of large language models as an adapt-retrieve-revise procedure. This approach combines the merits of efficiently performing continual training of a smaller 7B LLM for domain adaptation, robustly retrieving the supporting evidence from an external knowledge base, and effectively leveraging the evidence-assessing and revision capabilities of GPT-4. Our method significantly enhances the zero-shot performance of GPT-4 on four Chinese legal tasks.

While this paper manages to validate the effectiveness of the proposal in the Chinese legal domain, the adapt-retrieve-revise method itself is a flexible framework, which is expected to be adapted to a wide range of domains. We leave it as future work. Due to the substantial costs of the GPT-4 API, we could only sample a subset of test data during the evaluation. Resolving the trade-off between the growing experimental costs and the validity of evaluation remains a challenge for the GPT-4 research in the future.

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

# A  APPENDIX

## A.1  LEGAL INSTRUCTION TUNING

We build our legal instruction dataset by human experts. Due to privacy concerns, we are not allowed to disclose the annotated instruction data. However, we will release the instruction annotation guideline along with our 7B legal LLM. We show a template with a toy example below.

- **Due to the Article $x$ in the law $y$**: [the corresponding content in the law]
- **Considering the fact that** [the fact]
- **The judgment is** [the conclusion]

A toy example could be:

- **Due to the article 91 of the Road Traffic Safety Law of the People's Republic of China:** [Whoever drives a motor vehicle after drinking alcohol shall be imposed upon the penalty of temporary seizure of his motor vehicle driving license for not less than 1 month but not more than 3 months, and be imposed upon a fine of not less than 200 Yuan but not more than 500 Yuan as well; whoever drives a motor vehicle when he is drunk shall be restricted by the traffic administrative department of the public security organ until he becomes sober, be detained for not more than 15 days, be imposed upon the penalty of temporary seizure of his motor vehicle driving license for not less than 3 months but not more than 6 months, and be imposed upon a fine of not less than 500 Yuan but not more than 2000 Yuan as well. Whoever drives a commercial operating motor vehicle after drinking alcohol shall be imposed upon the penalty of temporary seizure of his motor vehicle driving license for 3 months, and be imposed upon a fine of 500 Yuan as well; whoever drives a commercial operating motor vehicle when he is drunk shall be restricted by the traffic administrative department of the public security organ until he becomes sober, be detained for not more than 15 days, be imposed upon the penalty of temporary seizure of his motor vehicle driving license for 6 months, be imposed upon a fine of 2000 Yuan as well. Where anyone is penalized for twice or more within one year due to his drunken driving as prescribed in the preceding two paragraphs, his motor vehicle driving license shall be canceled, and he shall not drive a commercial operating motor vehicle within 5 years.]
- **Considering the fact that** [the man was riding a motorbike when drunk.]
- **The judgment is** [to be restricted by the traffic administrative department of the public security organ until he becomes sober, be detained for not more than 15 days, be imposed upon the penalty of temporary seizure of his motor vehicle driving license for not less than 3 months but not more than 6 months, and be imposed upon a fine of not less than 500 Yuan but not more than 2000 Yuan as well.]

