# OpenReview forum: "Learning-Retrieval-Revision For Large Language Model Domain Adaptation"
_ICLR.cc/2024/Conference — Submitted to ICLR 2024_

### Official Review · Reviewer_CEqV · 2023-10-22

**Soundness:** 3 good
**Presentation:** 2 fair
**Contribution:** 2 fair
**Rating:** 5
**Confidence:** 4

**Summary:**

This paper studies the domain adaptation of large language models (LLMs) by focusing on the Chinese law domain. An adapt-retrieve-revise framework is proposed to adapt GPT-4 to the domain without modifying its own parameters. The initial step of the framework is to adapt a 7B LLM to the target domain through continual pre-training and the continual pre-trained model will be used to provide a draft answer. The framework will then use the draft answer to retrieve information and then call GPT-4 to revise the answer. The study shows that answer-based evidence retrieval yields much better results than question-based retrieval. The reported results also show that the proposed framework leads to much less hallucination and better recall on a series of tasks in the Chinese law domain.

**Strengths:**

1. The proposed framework is simple but effective. Tuning an affordable LLM to adapt GPT-4 can be viewed as another kind of parameter-efficient tuning. The draft output acts as a symbolic representation to link the trainable module and the fixed LLM. While many continual pre-training or continual learning methods cannot be applied to GPT-4, the work provides a general approach to use previous continual learning ideas to adapt GPT-4 to specific domains.
2. The experimental results are strong. Although this work is more empirical and does not have many theoretical contributions, the thorough analysis can bring interesting insights to the practitioners.

**Weaknesses:**

1. The presentation of this work needs improvement, especially the illustrations and their captions. Concise explanations should be added to the caption besides the caption title to make the illustrations more self-contained.
2. Some related works are missing. I think the paper should also discuss the literatures related to continual pre-training since it's a crucial part in the proposed adapt-retrieve-revise framework.
3. Regarding Table 1, I think one key experiment is missing: What's the result of using GPT-4 to provide a draft, retrieve with the draft answer, and revise the answer? I think the effect of continual pre-training is not adequately ablated in the current experiments.

**Questions:**

1. Could you provide the data statistics of datasets used in this work?
2. Is *recall* a commonly used metric on tasks LCR, CP, and LegalQA? I feel that with this metric alone, we cannot track whether the output contains some law causes that are related to the question. For example, the output can cover all the necessary law causes and also include some irrelevant law causes or hallucinate some non-existent law causes at the same time.

---

> ### Author Response · Authors · 2023-11-23
> **Thanks for your valuable suggestions and we provide more details in our respond.**
>
> Thanks for your valuable suggestions! We will provide more details to answer your questions.
>
> > Q1: The presentation of this work needs improvement, especially the illustrations and their captions. Concise explanations should be added to the caption besides the caption title to make the illustrations more self-contained.
>
> A1: Thanks for your suggestions, we will add concise descriptions into captions in the camera-ready version.
>
> > Q2: Some related works are missing. I think the paper should also discuss the literatures related to continual pre-training since it's a crucial part in the proposed adapt-retrieve-revise framework.
>
> A2: Thanks for your suggestions to improve the literatures. We decided to add related work in continual pre-training, and we so far list some papers to be added here:
> - Overcoming catastrophic forgetting in neural networks [Kirkpatrick +, 2016]
> - Continual learning in generative adversarial nets [Seff +, 2017]
> - icarl: Incremental classifier and representation learning [Rebuffi +, 2016]
> - Efficient meta lifelonglearning with limited memory [Wang +, 2020]
> - Overcoming catastrophic forgetting with hard attention to the task [Serrà +, 2018]
> - Supermasks in superposition [Wortsman +, 2020]
> - Continual learning in task-oriented dialogue systems [Madotto +, 2021]
> - Demix layers: Disentangling domains for modular language modeling [Gururangan +, 2021]
> - Continual training of language models for few-shot learning [Ke +, 2022]
> - ELLE: efficient lifelong pre-training for emerging data [Qin +, 2022]
> - LoRA: Low-Rank Adaptation of Large Language Models [Hu +, 2021]
> - Continual Pre-training of Language Models [Ke +, 2023]
>
> > Q3: Regarding Table 1, I think one key experiment is missing: What's the result of using GPT-4 to provide a draft, retrieve with the draft answer, and revise the answer? I think the effect of continual pre-training is not adequately ablated in the current experiments.
>
> A3: We already provide the result of using GPT-4 as both retrieval and reviser in Section 4.1.2. (GPT-4's answer-based retrieval in Figure 5). We agree that this result should be added into Table 1 and we will arrange it in the camera-ready stage. The effect of continual pre-training can be confirmed from both the zero-shot performance on each task (Table 1) and the ablation study on the quality of draft answers in retrieval (Figure 5).
>
> > Q4: Could you provide the data statistics of datasets used in this work?
>
> A4: Yes, we sampled a subset of #250 examples for the following four datasets due to the costs using GPT-4 API, and the following is the statistics of the original test set. We will add the statistics in the camera-ready stage.
>
> | Dataset | CP | LCR | LegalQA | JEC-QA |
> |---|---|---|---|---|
> |  #test | 965, 219  | 965, 219  | 1,000 | 13,341|
>
> > Q5: Is recall a commonly used metric on tasks LCR, CP, and LegalQA? I feel that with this metric alone, we cannot track whether the output contains some law causes that are related to the question. For example, the output can cover all the necessary law causes and also include some irrelevant law causes or hallucinate some non-existent law causes at the same time.
>
> A5: Thanks for your significant suggestion. Considering the fact that “the output can cover all the necessary law causes and also include some irrelevant law causes or hallucinate some non-existent law causes at the same time.”, we agree that recall alone will not reflect the true performance of each model. Thus, we updated the evaluation metric by adding F1 score into all results, and the following is the main result table with F1 scores.
>
> | Models | Retrieval | Revisor | LCR | CP | LegalQA | AVG. |
> |---|---|---|---|---|---|---|
> | Direct Generation |
> | GPT-4 | - | - | 61.67 | 70.72 | 11.97 | 48.12 |
> |7B Legal LLM | - | - | 82.99 | 82.85 | 41.91 | 69.25 |
> | Retrieval-based generation |
> | GPT-4 | Query-based | - | 72.04 | 74.01 | 33.1 | 59.72 |
> | 7B Legal LLM | Query-based | - | 77.32 | 81.3 | 47.36 | 68.66|
> |Adapt-Retrieve-Revise|
> | 7B Legal LLM | Answer-based | 7B Legal LLM | 84.1 | 82.72 | 47.25 | 71.36 |
> | 7B Legal LLM | Answer-based | GPT-4 | 90.57 | 86.87 | 71.08 | 82.84 |

---

### Official Review · Reviewer_kf7S · 2023-11-01

**Soundness:** 2 fair
**Presentation:** 2 fair
**Contribution:** 2 fair
**Rating:** 3
**Confidence:** 3

**Summary:**

This paper introduces a combination of continual training and retrieval-augmentation approaches to enhance GPT-4's performance in Chinese legal domain tasks. The proposed method consists of three steps: (1) first it adapts a 7B Chinese LM to the target domain using 50B Chinese legal data, (2) retrieves supporting evidence to a draft answer generated by the fine-tuned LM, and (3) feed the draft answers and evidence to GPT-4. Experimental results show improvements in Chinese legal QA datasets.
I have several major concerns about this paper: narrow problem focus and applicability to other languages or domains, limited novelties and soundness of evaluations, and presentations.

**Strengths:**

- This paper proposes a new method for domain adaption in Chinese legal domains, which conducts continual training of a Chinese LM on the target domain corpus and then retrieves relevant documents that suports or refutes the answers drafted by the smaller LM.
- By feeding the draft answer and retrieved evidence, proposed method obtains strong improvements on multiple Chinese legal QA datasets.

**Weaknesses:**

There are three main concerns about the papers about focus and motivations, soundness of evaluations, and the paper presentations.

**1. Narrow focus of the problem and wider applicability of the proposed method**

While improving the reliability of LMs in legal domains, especially more resource-constrained non-English language is important, this work focuses on a single language, and the proposed method heavily relies on the availability of 7B-size LMs in the target language and dozens of billions of domain corpus for continual pre-training. In many languages such rich resources are not available, and as the evaluations are mostly on Chinese legal domain datasets, it is unclear if the proposed method is applicable to other languages or not. If this work only focuses and improves zero-shot performance in Chinese legal domain tasks, it may not be interesting to wider audience of ICLR.

**2. Soundness of evaluations**

- **Evaluation metrics**

In Section 3.2, the authors mention that their evaluation simply relies on whether the answer includes the ground-truth law clause title is included or not.
> For tasks LCR, CP, and LegalQA, our metric is the recall of whether the title of the ground-truth law clause is included in the generated answer. This is because, in real-world applications, with the correct title, the contents of the law clause can be easily revised by the rule-based system, indicating that the title is more important than the content.

I don't think it is the proper way to evaluate accuracy or factuality of LM generation to legal domain related questions. For example, although this is an extreme case, if the only one metric is recall, a baseline always generating all clause titles can get perfect score. Using a recall of certain substrings to assess the quality of legal QA questions doesn't seem approppriate. Or if the end goal is to simply find a title of clause titles, I don't think using LM to generate full answers is the most optimal way to achieve such goal, and improving domain adapted retrieval system might be more suitable.
I checked the LegalQA dataset Github page, and their metrics seem to be MAP, MRR, and P@1, not recall. This makes me wonder why authors decided to use different metrics.
https://github.com/siatnlp/LegalQA

- **Baselines**
Given that the evaluation tasks are mostly evaluating whether a model can generate the correct title or a retrieval task, I think authors should include stronger retrieval models (at least multilingual encoder-based retrievals) as part of the pipelines or even as baselines to compare the proposed methods with. While authors claim that GPT-4with retrieval  (Query-based) is not as good as the proposed method, I suspect it is because their retrieval model is E5, which is not competitive compared to more recent embedding-based methods, and is not a multilingual retrieval system and the retrieval quality is poor. For instance, what happens if the authors use mContriever (Izacard et al., 2022) or even conduct continual pre-training of mContriever on the Chinese legal domain corpus used for 7B LM training?
Also except for JEC-QA, the evaluations are essentially retrieval tasks, so I wonder how well a competitive retrieval-based method perform on this task (given a question query, retrieves the clause text and title directly).

**3. Technical contributions or novelties of this work**

Continual training for domain adaptations or retrieval-augmentation for domain adaptations have been already studies. While drafting an answer from a small LM to search relevant document sounds somewhat new and interesting, due to the limited focus and evaluation protocol, I am not sure whether this can be widely applicable to other domains or indeed effective to enhance final generation quality.

**4. Presentation**

Occasionally, I found the paper is hard to follow, making it difficult to understand what the true contributions of this work is. Having a better structure might improve the paper presentations. For instance, Section 2 consists of low-level experimental details (the compuational requirements) as well as high-level ideas.

**Questions:**

- Why did you use recall as an evaluation metric?
- Did you try different retrieval methods, especially multilingual retrieval models?

---

> ### Author Response · Authors · 2023-11-23
> **Thanks for your valuable suggestions and we provide more details in our respond.**
>
> > Q1: Narrow focus of the problem and wider applicability of the proposed method
>
> A1: We appreciate this suggestion and are also interested in the gains in other domains. Unfortunately, the cost of extending experiments to another domain is beyond our capacity to afford. 7B-size LM is not a necessity in our proposed method but an anchor to use smaller and affordable LLMs to continually learn domain knowledge. In pracetice, you can leverage any existing in-domain models to achieve the adapt stage.
>
> However, from the research aspect, to ensure the reliability of experimental results, we need to perform continual training ourselves, instead of leveraging existing in-domain models as the first adapting step, of which the test data might be covered by their training data (widespread).  We will modify our paper title with (“-- a Case Study in Chinese Legal Domain”) to limit the scope.
>
> > Q2: Evaluation metrics
>
> A2: Thanks for your suggestion, and we provide an extra metric F1 in our paper. This link (https://github.com/siatnlp/LegalQA) is not the corresponding repository used in our paper. As mentioned in section 3.1, please refer to this repository (https://github.com/andongBlue/EQUALS) and the corresponding paper (https://dl.acm.org/doi/abs/10.1145/3594536.3595159). In their experiments, they used EM and F1 as their metrics, and their task was to make the span-level prediction extracted from the law articles, which is not a proper target in our case due to the nature of generative models. However, considering that “a baseline always generating all clause titles can get perfect score”, we agree that recall only will not reflect the actual performance of each model. Thus, we updated the evaluation metric by adding the F1 score to all results, and the following is the main result table with F1 scores.
>
> | Models | Retrieval | Revisor | LCR | CP | LegalQA | AVG. |
> |---|---|---|---|---|---|---|
> | Direct Generation |
> | GPT-4 | - | - | 61.67 | 70.72 | 11.97 | 48.12 |
> |7B Legal LLM | - | - | 82.99 | 82.85 | 41.91 | 69.25 |
> | Retrieval-based generation |
> | GPT-4 | Query-based | - | 72.04 | 74.01 | 33.1 | 59.72 |
> | 7B Legal LLM | Query-based | - | 77.32 | 81.3 | 47.36 | 68.66|
> |Adapt-Retrieve-Revise|
> | 7B Legal LLM | Answer-based | 7B Legal LLM | 84.1 | 82.72 | 47.25 | 71.36 |
> | 7B Legal LLM | Answer-based | GPT-4 | 90.57 | 86.87 | 71.08 | 82.84 |
>
> > Q3: Baselines
>
> A3: Although our main contribution does not stem from the retrieval component, we would agree with the importance of selecting strong baseline retrievers.
>
> For baseline methods, currently, multilingual E5-large is the SOTA family of text-embeddings which outperforms BM25, Contriever and GPT embeddings (Reference [1] and [2]).
> To further enhance our findings, we added extra experiments and compared with the current SOTA Chinese retrieval module CoROM following [3] on the LegalQA dataset, a similar setting as in Section 4.1.1 (Figure 4). Here are the results
> | Retriever  |  Retrieval |  top-1 | top-5  |  top-10 |
> |---|---|---|---|---|
> |  Multilingual E5-large | Query-based |45.8   | 61.5  |  70.3 |
> |  Multilingual E5-large | Answer-based | 65.3 |  84.5 |  88.5 |
> |  CoROM |  Query-based  | 47.5  |  60.8 |  71.5 |
> | CoROM | Answer-based | 58.8 | 72.5 | 80.5|
>
> From this table, we find that (1) multilingual e5-large has a competitive performance with CoROM on query-based retrieval, and largely outperforms CoROM on answer-based retrieval; (2) for both modules, the answer-based retrieval largely improves the retrieval quality than the query-based setting.
>
> References:
>
> [1]: Openai: <https://platform.openai.com/docs/guides/embeddings/what-are-embeddings>
>
> [2]: Text Embeddings by Weakly-Supervised Contrastive Pre-training [Wang + , 2022]
>
> [3]: DuReader-Retrieval: A Large-scale Chinese Benchmark for Passage Retrieval from Web Search Engine [Qiu +, 2022]
>
> > Q4: Technical contributions or novelties
>
> A4: Thanks for the expression of your concerns. Our proposal meets a very common demand; GPT-4 is not sufficiently good in some in-domain scenarios, but people lack the resources to fine-tune it. Our method does not heavily rely on any domain-specific modules. The initial adapting step can be accomplished by existing in-domain small models, which allows it to be applied to any other domains without significant adjustments. However, due to strict requirements for the reliability of results (e.g., test data should not be covered by training), we need to continually train 7B LLMs ourselves. The cost of experiments prevents us from verifying the effectiveness of our method in other domains. Nevertheless, we believe the flexibility of our framework for applying the method to other domains is unquestionable. We also added the F1 scores of the main results tables, which might somehow solve your concern about the evaluation protocol.
>
> > Q5: Presentation
>
> A5: Thanks for your suggestions, we will move the experimental details to Section 3, and also check similar issues in other parts.

---

### Official Review · Reviewer_TieX · 2023-11-01

**Soundness:** 2 fair
**Presentation:** 3 good
**Contribution:** 3 good
**Rating:** 5
**Confidence:** 4

**Summary:**

In this paper the authors proposed an adapt retrieve revise process for domain adaptation. They first train a model for certain domain and then use it to create a draft answer. This draft answer is used to retrieve evidence from some external knowledge base. Finally they use the draft answer and entire retrieved document ans query GPT 4 to revise the answer based on the evidence collected.

They conducted experiments on Chinese Legal corpora and show good improvements over the retrieval based generation.

**Strengths:**

The authors have contributed towards retrieval based generation and showed significant improvement over QA on certain domains like Chinese Legal QA.

This is an important area of research to improve the performance of LLM on low resource languages.

**Weaknesses:**

While this approach shows promise, it would be good to showcase if the sameis applicable or showing equivalent gains in other domains.
It would be good to run some ablation where using GPT4 used for retrieval as well as reviser. Also it would be interesting to understand how many evidence document is needed for a better revise mechanism. It seems that retrieval from the 7B model is zero shot, can that performance be improved ?

Is the revise a few shot generation? Also the author may discuss how just doing the revise of the draft based on the evidence have improved the recall so much in LCR, CP ad LegalQA.

The proposed approach seems to gain significant improvement over just retrieval based approach. Can the author provide some more details about what the retrieval mechanism used in the baseline? Also a comparison on the quality of retrieved evidence would be easier to understand.

Minor: The metrics etc should be mentioned in the table description for better understanding

**Questions:**

Please refer weaknesses,

---

> ### Author Response · Authors · 2023-11-23
> **Thanks for your valuable suggestions and we provide more details in our respond.**
>
> Thanks for your valuable suggestions! We will provide more details to answer your questions.
>
> > Q1: While this approach shows promise, it would be good to showcase if the same is applicable or showing equivalent gains in other domains. It would be good to run some ablation where using GPT4 used for retrieval as well as reviser. Also it would be interesting to understand how many evidence document is needed for a better revise mechanism. It seems that retrieval from the 7B model is zero shot, can that performance be improved ?
>
> A1: We already provide the result of using GPT-4 as both retrieval and reviser in Section 4.1.2. (GPT-4's answer-based retrieval in Figure 5). The results show that the poor quality of GPT-4’s answers leads to noisy retrieval and inaccurate revision compared with using the draft answer of the 7B legal LLM in retrieval.
>
> We appreciate this suggestion and are also interested in the gains in other domains. Unfortunately, the cost of extending experiments to another domain is beyond our capacity to afford. To ensure the reliability of experimental results, we need to perform continual training ourselves, instead of leveraging existing in-domain models as the first adapting step, of which the test data might be covered by their training data (widespread). We will modify our paper title (“-- a Case Study in Chinese Legal Domain”) to limit the scope.
>
> We are also interested in the number of documents needed in revise mechanism. Qualitatively, we believe evidence documents are not the more the better, but should rely on the corresponding database for the target task. However, we have not implemented quantitative experiments yet and will leave it as future work.
>
> > Q2: Is the revise a few shot generation? Also the author may discuss how just doing the revise of the draft based on the evidence have improved the recall so much in LCR, CP and LegalQA.
>
> A2: All generation in this paper is zero-shot. We concluded the reason to be (1) the accurate draft answer by legal LLMs to retrieve related evidence and (2) the evidence-assessing capability of GPT-4 to revise the draft answer referring to evidence. We provided three types of improvements in the case study section 4.2.
>
> > Q3: The proposed approach seems to gain significant improvement over just retrieval based approach. Can the author provide some more details about what the retrieval mechanism used in the baseline? Also a comparison on the quality of retrieved evidence would be easier to understand.
>
> A3: For the retrieval mechanism used in baseline methods, we also used multilingual E5-large, which is the SOTA family of text-embeddings and outperforms BM25, Contriever, and GPT embeddings (Reference [1] and [2]).
>
> To further enhance our findings, we added extra experiments and compared them with the current SOTA Chinese retrieval module CoROM following [3] on the LegalQA dataset, the same setting as in Section 4.1.1 (Figure 4). Here are the results
> | Retriever  |  Retrieval |  top-1 | top-5  |  top-10 |
> |---|---|---|---|---|
> |  Multilingual E5-large | Query-based |45.8   | 61.5  |  70.3 |
> |  Multilingual E5-large | Answer-based | 65.3 |  84.5 |  88.5 |
> |  CoROM |  Query-based  | 47.5  |  60.8 |  71.5 |
> | CoROM | Answer-based | 58.8 | 72.5 | 80.5|
>
> From this table, we find that (1) multilingual e5-large has a competitive performance with CoROM on query-based retrieval, and vastly outperforms CoROM on answer-based retrieval; (2) for both modules, the answer-based retrieval largely improves the retrieval quality than the query-based setting.
>
> References:
>
> [1]: Openai: <https://platform.openai.com/docs/guides/embeddings/what-are-embeddings>
>
> [2]: Text Embeddings by Weakly-Supervised Contrastive Pre-training [Wang +, 2022]
>
> [3]: DuReader-Retrieval: A Large-scale Chinese Benchmark for Passage Retrieval from Web Search Engine [Qiu +, 2022]
>
> > Q4: Minor: The metrics etc should be mentioned in the table description for better understanding
>
> A4: Thanks for your suggestions for improving our paper. We will add more concise descriptions for each table and figure.

---

### Official Review · Reviewer_n3Gy · 2023-11-04

**Soundness:** 3 good
**Presentation:** 2 fair
**Contribution:** 3 good
**Rating:** 6
**Confidence:** 4

**Summary:**

This paper introduces a domain adaptation framework for LLMs, reimagining generation as a three-step adapt-retrieve-revise process. The authors present a straightforward yet effective technique for adapting a smaller LLM to a specific target domain through continued learning on in-domain data. Subsequently, the adapted LLM is employed to generate an initial draft in response to a task query. This draft is then used to achieve more precise retrieval compared to using the query alone. The proposed method has been shown to significantly enhance the accuracy of LLMs in knowledge-intensive domains (i.e., legal domain).

**Strengths:**

1. This paper investigates an important research question: how to adapt LLMs (e.g., GPT-*) to knowledge-intensive domains. Their proposed pipeline is simple yet effective for solving tasks within the legal domain.
2. The high-level idea of using candidate answers to create a more informative query for improving retrieval performance is novel.
3. The authors conduct comprehensive experiments and ablation studies across several legal tasks to demonstrate the effectiveness of the proposed pipeline.

**Weaknesses:**

1. More established retrieval modules (e.g., BM25, Contriever, or GPT embeddings) should be investigated to enhance the robustness of the findings.
2. The study is limited to GPT-4 models and explores only a subset of each task. The authors should consider including other open-source LLMs, such as Llama-2, to demonstrate the generalizability of the proposed methods.

**Questions:**

In real-world scenarios, training domain-specific smaller LLMs may still be computationally intensive and time-consuming. Is there a way to mitigate these challenges without significantly compromising the model's performance that can be generally extended to other knowledge-intensive domains (e.g., medical, financial)?

---

> ### Author Response · Authors · 2023-11-23
> **Appreciate for suggestions and add more details in the response**
>
> Thanks for your valuable suggestions! We will provide more details to answer your questions.
>
> > Q1: More established retrieval modules (e.g., BM25, Contriever, or GPT embeddings) should be investigated to enhance the robustness of the findings.
>
> A1: Thanks for your suggestions! In this paper, we leveraged multilingual E5-large which is the SOTA family of text-embeddings, which has been reported to outperforms BM25, Contriever and GPT embeddings (Reference [1] and [2]). Since the improvements of our method are consistent and substantial (+33.3% vs vanilla, +15.4%/23.9% vs retrieval baselines), we believe these gaps have shown sufficient robustness of our proposal.
>
> However, we agree that including more established retrieval modules can enhance the robustness of our findings. Therefore, we added extra experiments and compared with the current SOTA chinese retrieval module CoROM following [3] on LegalQA dataset, the same setting as in Section 4.1.1 (Figure 4). Here are the results
> | Retriever  |  Retrieval |  top-1 | top-5  |  top-10 |
> |---|---|---|---|---|
> |  Multilingual E5-large | Query-based |45.8   | 61.5  |  70.3 |
> |  Multilingual E5-large | Answer-based | 65.3 |  84.5 |  88.5 |
> |  CoROM |  Query-based  | 47.5  |  60.8 |  71.5 |
> | CoROM | Answer-based | 58.8 | 72.5 | 80.5|
>
> From this table, we find that (1) multilingual e5-large has a competitive performance with CoROM on query-based retrieval, and vastly outperforms CoROM on answer-based retrieval; (2) for both modules, the answer-based retrieval primarily improves the retrieval quality than the query-based setting.
>
> References:
>
> [1]: Openai: <https://platform.openai.com/docs/guides/embeddings/what-are-embeddings>
>
> [2]: Text Embeddings by Weakly-Supervised Contrastive Pre-training [Wang +, 2022]
>
> [3]: DuReader-Retrieval: A Large-scale Chinese Benchmark for Passage Retrieval from Web Search Engine [Qiu +, 2022]
>
> > Q2: The study is limited to GPT-4 models and explores only a subset of each task. The authors should consider including other open-source LLMs, such as Llama-2, to demonstrate the generalizability of the proposed methods.
>
> A2: We agree that more open-source LLMs would enhance the generalizability of our method. In our experiments, we have used the open-source 7B LLM in the revision stage (Table 1) and argue that 7B LLM does not have the evidence-assessing capability for the revision. However, for 65B Llama-2, due to the weak capacity of Chinese (compared to GPT4) and the substantial computational cost, it is not an optimal choice. At the same time, GPT-4 APIs could be widely used and provide the current best performance.
>
> > Q3: In real-world scenarios, training domain-specific smaller LLMs may still be computationally intensive and time-consuming. Is there a way to mitigate these challenges without significantly compromising the model's performance that can be generally extended to other knowledge-intensive domains (e.g., medical, financial)?
>
> A3: Our proposed method is a flexible framework, and any existing in-domain models can replace the first adapt stage. For example, we have introduced plenty of Chinese legal LLMs in the related work, which can be further integrated into our adapt-retrieve-revise framework. To clarify, the reason that we did not include these existing models in our experiments is that we can not estimate whether the test data is covered by the training data of the current models, which would influence the reliability of our experiment results (mentioned in Section 5.2). Thus, in other knowledge-intensive domains, one trial way is to integrate a current in-domain model into our adapt-retrieve-revise framework to check the improvement. The other way is that we can use supervised fine-tuning to train LLMs on some domain datasets without high computational costs and test the performance of this adapt-retrieve-revise method with other existing models.

---

### Meta-Review · Area_Chair_CyWK · 2023-12-06

**Metareview:**

Domain adaptation for LLMs is a pressing issue - the paper tackles an important problem.
The proposed approach has merit -  using ChatGPT to revise the answers of domain-tuned smaller models.
Despite its strengths,  the paper has several weaknesses regarding: reliability of evaluation, limited comparison to established retrieval methods, a focus only on GPT-4 models ignoring other LLMs, and a narrow scope restricted to the Chinese legal domain. The last part is not such a big problem given that many published papers also focus on just English, and a single task domain.  There are also some presentation issues.
We suggest the authors focus on improving their evaluation, and extension of the their approach to broader LLM concepts and tools, and the paper's presentation.

**Justification For Why Not Higher Score:**

Despite its strengths,  the paper has several weaknesses regarding: reliability of evaluation, limited comparison to established retrieval methods, a focus only on GPT-4 models ignoring other LLMs, and a narrow scope restricted to the Chinese legal domain. Those the last part is not such a big problem given that many papers also focus on just English, and a single task (movie reviews).  There are also some presentation issues.

**Justification For Why Not Lower Score:**

N/A

---

### Decision · Program_Chairs · 2024-01-16

Reject